# Sex, Age, and Previous Herpes Zoster Infection Role on Adverse Events Following Immunization with Adjuvanted Recombinant Vaccine

**DOI:** 10.3390/pathogens14020195

**Published:** 2025-02-15

**Authors:** Maria Costantino, Valentina Giudice, Giuseppina Moccia, Monica Ragozzino, Salvatore Calabrese, Francesco Caiazzo, Massimo Beatrice, Walter Longanella, Simona Caruccio, Candida Iacuzzo, Carmen Giugliano, Mariagrazia Bathilde Marongiu, Giovanni Genovese, Bianca Serio, Emilia Anna Vozzella, Amelia Filippelli, Francesco De Caro

**Affiliations:** 1Department of Medicine, Surgery, and Dentistry, University of Salerno, 84081 Baronissi, Italy; vgiudice@unisa.it (V.G.); gmoccia@unisa.it (G.M.); scaruccio@unisa.it (S.C.); afilippelli@unisa.it (A.F.); fdecaro@unisa.it (F.D.C.); 2University Hospital “San Giovanni di Dio e Ruggi d’Aragona”, 84131 Salerno, Italy; monica.ragozzino@sangiovannieruggi.it (M.R.); salvatore.calabrese@sangiovannieruggi.it (S.C.); francesco.caiazzo@sangiovannieruggi.it (F.C.); massimo.beatrice@sangiovannieruggi.it (M.B.); walter.longanella@sangiovannieruggi.it (W.L.); candida.iacuzzo@sangiovannieruggi.it (C.I.); carmen.giugliano@sangiovannieruggi.it (C.G.); giovanni.genovese@sangiovannieruggi.it (G.G.); bianca.serio@sangiovannieruggi.it (B.S.);; 3Department of Women, Child and General and Specialized Surgery, University “Luigi Vanvitelli”, 80138 Naples, Italy

**Keywords:** recombinant zoster vaccine (RZV), gender difference, age-related responses, AEFI, fragile populations

## Abstract

Adverse events following immunizations (AEFIs) with recombinant zoster vaccine (RZV) are underexplored in fragile populations. This study aims to assess incidence, duration, and characteristics of AEFIs, focusing on the impact of sex, age, and prior Herpes Zoster (HZ) infection in a frail population, including solid organ transplant recipients. We conducted an observational study on patients receiving RZV, and AEFIs were classified as local or systemic and analyzed for incidence, duration, and patterns across groups. We showed that females had a higher incidence of AEFIs (*p* = 0.02), both local and systemic symptoms, such as swelling +/− redness at the site of injection and fatigue, after the first and second doses. Younger adults experienced more systemic reactions, while older adults reported more local events (e.g., redness and swelling, *p* = 0.01). Moreover, patients with previous HZ infection exhibited a higher incidence of AEFIs after the second dose (68% vs. 38%, *p* = 0.001). In conclusion, sex, age, and clinical history significantly influenced AEFI incidence and manifestations. Therefore, it is important to personalize vaccination strategies in frail populations, by tailored administration and monitoring plans, especially in females and individuals with prior HZ infection, to improve vaccine safety and patient outcomes.

## 1. Introduction

Vaccinations are one of the most effective measures for infectious disease prevention and constitute a key strategy for combating the development and spread of antibiotic resistance [1,2,3,4]. Adjuvanted recombinant vaccines are of great interest in public health because of their ability to enhance the immunogenicity of vaccine antigens, and this characteristic is particularly important in subjects with reduced immune system functions, such as the elderly and immunocompromised individuals, like solid organ transplant recipients [5].

Vaccines could be associated with adverse events, known as adverse events following immunizations (AEFIs), and their incidence and severity can vary depending on individual factors, including sex and age [6,7,8]. Indeed, females exhibit stronger immune responses to vaccines compared to males, often producing two-fold increased antibody titers [6,9]. However, this enhanced immunoreactivity is associated with more frequent and severe AEFIs, as also documented by the Italian Medicine Agency AIFA in its annual report of vaccine post-marketing [10,11]. Various factors can contribute to these differences, such as sex hormone-related changes in immune system composition and functions, favoring T helper 1 (Th1) and B cell-mediated responses in females, while males have enhanced innate responses and pro-inflammatory cytokines levels [12,13,14,15,16]. Moreover, females are more likely to report AEFIs than males [9]. Despite this evidence, sex-based differences in immune responses are still not adequately considered in study design and dose findings of drugs and vaccines, and AEFI incidence, distribution, and risk factors could be under-estimates and data remain limited [5,11]. Age also plays a crucial role in modulating immune responses to vaccines [13], as immunosenescence is frequent in older adults, causing attenuated responses to pathogens and vaccines, as well as AEFI occurrence [17].

Herpes Zoster (HZ) is a disease caused by the reactivation of the varicella-zoster virus (VZV), a Herpes virus that remains latent in the dorsal root ganglia and cranial nerves after primary infection [18]. Viral reactivation is favored by a decline in cell-mediated immunity due to immunosenescence or immunosuppression, such as during acquired or congenital immunodeficiency syndromes or under immunosuppressive therapies. HZ manifests as painful, often unilateral vesicular skin eruptions, and can lead to severe acute complications and/or to long-term syndromes, such as postherpetic neuralgia, negatively affecting quality of life [19]. Therefore, vaccination is a necessary preventive measure to reduce both incidence and severity of HZ and its complications, especially in vulnerable populations. In the elderly, HZ infection is frequently associated with a high rate of hospitalization, up to six-fold compared to younger subjects (up to 25 hospitalizations/100,000 patient/years), and an immunocompromised status can increase this rate, with an in-patient mortality up to two-fold [20]. For example, diabetic patients have a hospitalization rate of up to 15.9 per 100,000, and those subjects aged 65 or older, with a BMI > 30, and poor glycemic control (HbA1c > 8.0%) are even more at risk of rehospitalization (40%) and of post-HZ complications (25% higher risk) [21].

Currently, an inactivated adjuvanted recombinant vaccine against Herpes Zoster (RVZ) is available, providing a high level of protection in the general population and fragile individuals [22]. Although several studies have explored the impact of age and sex on immune responses after vaccinations, detailed characterization of AEFI occurrence and severity in vulnerable populations after adjuvanted recombinant vaccines remains an under-researched area. In this retrospective real-life study, we aimed to analyze incidence and types of AEFIs following RVZ immunization in vulnerable populations, especially in those subjects with prior HZ manifestations. A better understanding of risk factors of AEFI development could optimize and personalize strategies, improving vaccination safety and acceptance.

## 2. Materials and Methods

### 2.1. Study Design

In this observational real-life study, a total of 174 frail Caucasian adults (mean age, 57 ± 11.8 years; range, 28–84 years) who received RZV at the “Ruggi” hospital site, University Hospital “San Giovanni di Dio e Ruggi d’Aragona”, Salerno, Italy, between 2022 and 2024 were included. The RZV schedule consisted of two intramuscular doses of 0.5 mL each, with an interval of 2 to 6 months between each administration, as per Italian guidelines and technical datasheet. Vaccination was free of charge as per clinical Italian practice and was administered at the site hospital Ruggi following the Italian Health Institute’s recommendations. No other vaccinations were administered together with RZV, and at least one month elapsed between different vaccine doses. After vaccination, patients were monitored by periodical out-patient visits. The inclusion criteria were as follows: age ≥ 18 years old; presence of severe immunodeficiency conditions, such as hematological malignances, hematopoietic stem cell or solid organ transplantation, or human immunodeficiency virus (HIV) infection; the presence of geriatric frailty, defined as individuals aged 65 years or older with reduced physiological reserve and resilience; the presence of comorbidities that posed an indication for RZV, including type 2 diabetes, obesity, or autoimmune diseases; the completion of the vaccination cycle; and signed informed consent. The exclusion criteria were the following: no indications for RZV; incomplete vaccination; and refusal to sign informed consent. Clinical characteristics are summarized in Table 1.

This study followed the Declaration of Helsinki and was approved by local Ethics Committee “Campania Sud”, Naples, Italy (n. 185_r.p.s.o./2022).

### 2.2. Demographic and AEFI Data Collection

A specific Case Report Form (CRF) was used to collect patients’ data, including underlying conditions and previous HZ disease, and type, duration, and treatments used for AEFIs and severe AEFIs (SAEFIs) after first and second dose administration. AEFIs were classified according to the Italian and European guidelines [23,24].

### 2.3. Statistical Analysis

Data were collected in a spreadsheet and analyzed using SPSS 23.0 statistics package. Continuous variables were presented as mean ± standard deviation (SD). Two-group comparisons were performed using an unpaired or paired *t*-test for normally distributed data, while categorical variables were analyzed using Chi square test or Fisher’s exact test for multiple groups. A logistic regression analysis was performed to investigate the effects of several factors on AEFI occurrence. A *p* value < 0.05 was considered statistically significant.

## 3. Results

### 3.1. Females More Frequently Report Mild/Moderate AEFIs After First RZV Dose

To explore sex-related differences in AEFI incidence, types, and duration, patients were stratified by sex (females, N = 51, 29%; and males, N = 123, 71%). In both groups, kidney transplant recipients were the most represented (76% females, and 70.7% males), while HIV infection was exclusively present in males (10.6%), as well as geriatric frailty (4.1%) (Table 2). As expected, females more frequently complained of AEFIs after the first dose of RZV (N = 28; 55%) compared to males (N = 44; 35.8%) (*p* = 0.02), without differences in duration (75 ± 74.8 h vs. 66 ± 114.8 h, females vs. males; *p* = 0.7), either as a local or systemic event (all *p* > 0.05) (Table 2). Paracetamol was used to treat AEFIs (100% in both females and males). Conversely, local and systemic AEFIs showed a variable distribution between sexes (Figure 1). In detail, females more frequently reported local AEFIs after the first RZV dose compared to males, especially swelling and/or redness at the injection site (14% vs. 2%, females vs. males; *p* = 0.0036), while males were more likely to experience pain at the injection site (77% vs. 64%, males vs. females; *p* = 0.088). For systemic AEFIs, females more commonly reported fatigue compared to males (29% vs. 16%, females vs. males; *p* = 0.0488) (Figure 1A).

After the second vaccine dose, males presented more AEFIs than those observed after the first dose (N = 50; 41%); however, females tended to display adverse events more frequently (N = 28; 55%) (*p* = 0.09). Total symptom duration was similar in both sexes (51 ± 40.4 h vs. 48 ± 42.7 h, females vs. males; *p* = 0.8), regardless of being local or systemic (all *p* > 0.05) (Table 2). Paracetamol was used to treat AEFIs (100% in both females and males). As observed for AEFI distribution after the first dose, local and systemic symptoms showed a variable distribution between the sexes. In particular, females more frequently reported local AEFIs after the second RZV dose compared to males, especially swelling and/or redness at the injection site (24% vs. 2%, females vs. males; *p* < 0.0001) and itching (4% vs. 0%, females vs. males; *p* = 0.0272). For systemic AEFIs, females more commonly experienced muscle/joint pain after the second dose compared to males (28% vs. 6%, females vs. males; *p* = 0.0002) (Figure 1B). No significant variations between sexes for the remaining symptoms were observed (all *p* > 0.05).

### 3.2. Young Adults More Frequently Report Local and Systemic AEFIs

Next, to investigate the effects of age on AEFI development after RZV, patients were divided in three groups based on age at vaccination: young adults aged 28–39 years; adults aged 40–65 years; and older adults aged > 65 years. After the administration of the first dose, total AEFI incidence tended to progressively decrease with age, as older adults complained of an AEFI only in 30% of cases compared to 54% in young adults and 44% in adults (all *p* > 0.05). This trend could be explained with the phenomenon of immunosenescence in older adults, that could also be related to the lower incidence of both systemic and local adverse events (17% and 26%, respectively; all *p* > 0.05) (Table 3). Total duration did not significantly differ between age groups, either as local or systemic symptoms (all *p* > 0.05). Based on manifestations, young adults tended to more frequently complain of local (54% vs. 35% vs. 26%; *p* = 0.11) and systemic AEFIs (38% vs. 27% vs. 17%; *p* = 0.21) compared to adults and elderly. In particular, redness and/or swelling at the injection site was significantly more frequent in young adults compared to adults and elderly (23% vs. 1% vs. 2%, respectively; *p* = 0.004). No other significant differences were observed between age groups in types of AEFI manifestations.

Similarly, after the administration of the second dose, total AEFI incidence was significantly higher in younger subjects, as older adults reported an AEFI only in 37% of cases compared to 69% in young adults and 45% in adults (*p* = 0.04), with lower incidence of both systemic and local adverse events (37% and 26%, respectively; *p* = 0.11 and *p* = 0.12) (Table 3). Total duration did not significantly differ between age groups, either as local or systemic symptoms (all *p* > 0.05), as observed for AEFIs after the first dose. Based on manifestations, young adults tended to more frequently complain of local (54% vs. 37% vs. 26%; *p* = 0.11) and systemic AEFIs (69% vs. 42% vs. 37%; *p* = 0.12) compared to adults and elderly. No significant differences were observed for local AEFIs, while young adults more commonly reported fever or flu-like syndrome compared to adults and elderly (55% vs. 42% vs. 30%, respectively; *p* = 0.04). No other significant differences were observed between age groups in types of AEFI manifestations, as well as duration of local and systemic AEFIs (Table 3).

### 3.3. Previous AEFI Occurrence Does Not Increase Severity and Incidence of Adverse Events After Second RZV Dose

Subsequently, we thought to study the safety of RZV in subjects who have experienced an AEFI after the first dose, and we investigated incidence, type, and duration of a second AEFI in this subgroup of patients. Among all 174 vaccinated subjects, 49 of them (28%) after both the first and second doses, while 31 patients (18%) reported an AEFI only after the second administration, and 71 (41%) reported no cases. To explore incidence and severity of AEFIs during the second dose, we compared subjects who experienced an AEFI after both the first and second administration (Group I/II AEFI) with those who reported an AEFI only after the second dose (Group II AEFI) (Table 4).

Group I/II AEFI tended to exhibit a higher incidence of systemic AEFIs (96%) and lower local symptoms (69%) compared to Group II AEFI (84% and 87%, respectively; *p* = 0.06 and *p* = 0.07). These findings suggested that patients who experience an AEFI after the first dose could develop a more robust (and potentially more reactogenic) immune response after the second dose. For specific symptoms, injection site pain was the most common local AEFI, occurring more frequently in Group II (77%) than in Group I/II (61%), although the difference did not reach statistical significance (*p* = 0.1). Among systemic AEFI, the most frequent symptoms included fever, fatigue, and headache, with similar distributions between the two groups (Table 4). However, no differences were observed between groups in terms of total duration, local and systemic AEFI persistence, and types of manifestations (all *p* > 0.05). After the second dose, medications to alleviate AEFI symptoms were more commonly employed in Group I/II AEFI compared to Group II AEFI (35% vs. 16%; *p* = 0.07), and paracetamol was the only drug used (Table 4).

### 3.4. Impact of Previous Herpes Zoster Infection on Prevalence and Duration of AEFI

Finally, we investigated the efficacy and safety of RZV in patients who had a clinical history of HZ and the impact of this vaccination on reducing viral reactivation. For this purpose, patients were stratified based on previous HZ infection, and 78% of subjects (N = 136) reported no prior HZ, while 22% of the total cohort (N = 38) had a clinical history of HZ. However, none of these subjects reported HZ infection in the 12 months before the vaccination. Following the first dose, overall AEFI incidence was similar between groups (40% vs. 45%, without and with previous HZ; *p* = 0.6), indicating that a previous HZ infection could be not related to AEFI development after RZV. However, once experienced, patients with a clinical history of HZ were more likely to have systemic symptoms (82% vs. 56%, with and without previous HZ; *p* = 0.05) (Table 5). No differences were observed between groups in terms of total duration, local and systemic AEFI persistence, and types of manifestations (all *p* > 0.05). After the second dose, the incidence of AEFIs was significantly higher in subjects with previous HZ compared to those without a clinical history (68% vs. 38%, respectively; *p* = 0.001) (Table 5). This finding suggested that patients with a previous HZ infection could have a greater antigen susceptibility and an enhanced immune response, likely because the second vaccine dose could act as a “third” booster to their immune system. No differences were observed between groups in terms of total duration, local and systemic AEFI persistence, and types of manifestations (all *p* > 0.05).

Finally, we confirmed the effects of these variables on the occurrence of AEFIs after the first and second RZV doses by multivariate logistic regression analysis (Table 6). In particular, female sex (*p* = 0.0415) and the presence of other comorbidities (*p* = 0.0194), including rheumatological disorders, hematological malignancies, diabetes, or immunological diseases, were significantly associated with the occurrence of AEFIs after the first vaccine dose. Conversely, age (*p* = 0.0300), clinical history of HZ infection (*p* = 0.0378), regardless of the number of previous reactivations (*p* = 0.5100), and the occurrence of AEFIs after the first dose (*p* = 0.0007) were significantly associated with the development of AEFIs after the second RZV administration.

## 4. Discussion

Vaccination is the most important preventive measure in public health to reduce infectious disease-related mortality and morbidity worldwide [25]. HZ results from the reactivation of VZV, which is latent in the sensory ganglia after primary infection [26]. Viral reactivations are usually caused by reduced immunosurveillance due to impaired cellular-mediated immunity related to aging, immunosenescence, and immunosuppressive status [27]. Therefore, HZ can frequently occur in frail populations, and its related complications, such as post-herpetic neuralgia, can significantly impact the quality of life of these subjects [28]. The individual lifetime risk of developing HZ increases with age, especially after 50 years of age with two-thirds of all cases, and post-herpetic neuralgia can persist for more than 1 year in 10% of affected subjects, mainly the elderly (60–70% of all cases of post-HZ neuralgia) [29]. Moreover, HZ-related mortality is often under-considered. It poses a public health problem, as in Europe, the median mortality incidence in those aged ≥ 50 years is 0.039 per 100 000, and is particularly higher in females, although in some European Countries, such as Spain, Italy, France, and Denmark, males are more affected [30]. HZ-related mortality also increases with aging, with higher incidence in subjects aged 70–74 years old [27]. Therefore, HZ vaccination can dramatically abolish mortality and morbidity in general and frail populations.

In our previous investigation, we confirmed the efficacy and safety of this novel RZV in a real-life setting, comprising immunocompromised subjects, especially solid organ and hematopoietic stem cell transplant recipients [22]. Indeed, the AEFI occurrence rate is similar to that reported in clinical trials and in immunocompetent subjects, and events are usually of mild or moderate severity, requiring local treatments or nonsteroidal anti-inflammatory drugs [31,32,33]. In this retrospective real-life study, we confirmed and expanded the safety of RZV in a frail population, and we showed that females were more susceptible to developing AEFIs compared to males, especially local symptoms after the first dose and joint and/or muscle pain after the second administration. This higher AEFI incidence in females after RZV is in accordance with published data on other vaccinations, as females have a more active cellular-mediated immune response compared to males, and this is linked to common and/or severe AEFI occurrence, as also documented annually by local and international medicine agencies [10]. For example, in Italy, AEFIs after COVID-19 immunization were predominantly reported by females (76%), as well as in the US (females, 78.7% of total reported side effects, by US-CDC reports) [34]. This discrepancy could be related to an increased attitude of females to report mild to moderate side effects; however, severe adverse reactions, that are principally reported by healthcare professionals, such as anaphylaxis, are more common in females (80% of cases from 1990 to 2016 by CDC reports) [35]. Therefore, our results confirmed the general observations of increased incidence of AEFIs in females after any vaccination [6,9,22,36,37,38,39,40]. For example, females are at an increased risk of injection site and systemic reactions following influenza vaccines compared to males, regardless of age [37], as well as anaphylaxis, occurring in females in up to 83% of cases after various vaccinations [39], or immediate hypersensitivity after administration of the monovalent 2009 pandemic influenza A (H1N1) vaccine, that is four times more common in females, especially in those aged 30–39 years old [40]. Moreover, after at least one dose of the AZD1222 vaccine against SARS-CoV-2, the most common AEFIs are moderate fever (69.4%), muscle aches (68.6%), fatigue/sleepiness (62.5%), body aches (59.4%), headache (58.5%), pain at the injection site (58.3%), and chills (45.7%), and females are more likely to report gastrointestinal symptoms and other AEFIs compared to males [41]. Similarly, in our cohort receiving RZV, the most commonly reported AEFIs were fever, pain at the injection site, and fatigue, especially in females.

Sex-related differences in immune responses are principally caused by sex hormones and genetic/epigenetic mechanisms that modulate immune system composition and functions [12,42,43,44]. For example, seasonal TIV with a half dose of vaccine can induce protective antibody titers in females, with similar levels observed in males after a full dose vaccination [45]. Moreover, mRNA-based vaccines seem to be more effective in males compared to females, although AEFIs can occur more frequently in women [46]. Despite these known differences, design and dose finding studies do not consider sex-related immune system variations. For RZV, in a phase IV study, 4381 adverse events were reported in the first eight months of marketing, with fever (23.6% of cases) and pain at the injection site (22.5%) occurring more frequently [47]. A similar safety profile was also observed in patients with rheumatoid arthritis under biologic agents or JAK inhibitors [48]. However, no results stratified by sex have been reported yet [48]. Here, we showed that females were more likely to report local and systemic AEFIs following the first and second RZV dose compared to males, especially swelling and/or redness at the injection site, fatigue, and joint and/or muscle pain, while males were more likely to report pain at the injection site, especially after the first dose.

Age disparities in AEFI occurrence and severity were reported for influenza and COVID-19 vaccines, with young adults and adults more frequently reporting AEFIs, while older adults aged 65 years or older have a higher incidence of severe side effects and deaths following either quadrivalent virus-like particle vaccination or quadrivalent inactivated vaccination [49,50]. Our results are consistent with age and gender disparities in AEFIs, as our young adults more commonly reported side effects compared to the elderly, especially systemic symptoms. These data emphasized the importance of adopting targeted monitoring strategies by age groups, preparing young individuals for a higher incidence of systemic symptoms, and implementing preventive measures to reduce discomfort in older adults.

Finally, we showed that patients with a clinical history of HZ infection had an increased incidence of AEFIs, particularly after the second dose of the vaccine, likely because of enhanced immune reactivity mediated by antigenic memory stimulated by the vaccine. However, duration and clinical manifestations were similar to those observed in patients without a clinical history of HZ. This observation could be of importance in vulnerable subgroups, such as patients with comorbidities or frailty and a prior HZ infection, to better plan a vaccine schedule, dosage, and monitoring.

Our study has some limitations: (i) the observational nature of the designed investigation and inclusion of a relatively small sample size with some biases related to disease-specific prevalence (e.g., HIV infection was predominant in males, while kidney transplant recipients were females); (ii) a short follow-up for the identification of long-term vaccine-related complications; and (iii) the lack of a control group without comorbidities and other immunocompromised conditions.

## 5. Conclusions

In conclusion, incidence and characteristics of AEFIs after RZV can be influenced by several factors, such as sex, age, and previous HZ infection. Indeed, sex-related differences in immune responses are caused by hormone effects on immune cell maturation and distribution, favoring innate responses and inflammation in males and adaptive immunity in females. This physiological diversity is associated with a higher risk of severe infectious diseases in males compared to females, while they show a better response to vaccination with a lower incidence of AEFIs. Moreover, age also modifies immune system composition and activation, as immunosenescence frequently occurs with aging, making immune responses less efficient. In our study, we confirmed that sex and age are principal risk factors of AEFIs in females after RZV, and younger subjects have less pronounced immune system activation compared to the elderly. In addition, individuals with a history of HZ infection could exhibit altered immune responses due to prior sensitization, which could influence the severity and frequency of AEFIs. Our study emphasizes the importance of considering sex, age, and clinical history when planning a vaccine schedule, dosage, and monitoring, particularly in vulnerable populations. Differences in incidence and types of AEFIs between sexes and age groups highlight the need to tailor vaccine administration, to optimize its efficacy and minimize discomfort and side effects. A targeted focus on frail populations, for whom currently available data are limited, could significantly contribute to improving vaccination planning and AEFI management. Future studies on larger cohorts stratified by clinical and demographic characteristics will be crucial to confirm our observations and to identify potential predictive factors of AEFI occurrence.

## Figures and Tables

**Figure 1 pathogens-14-00195-f001:**
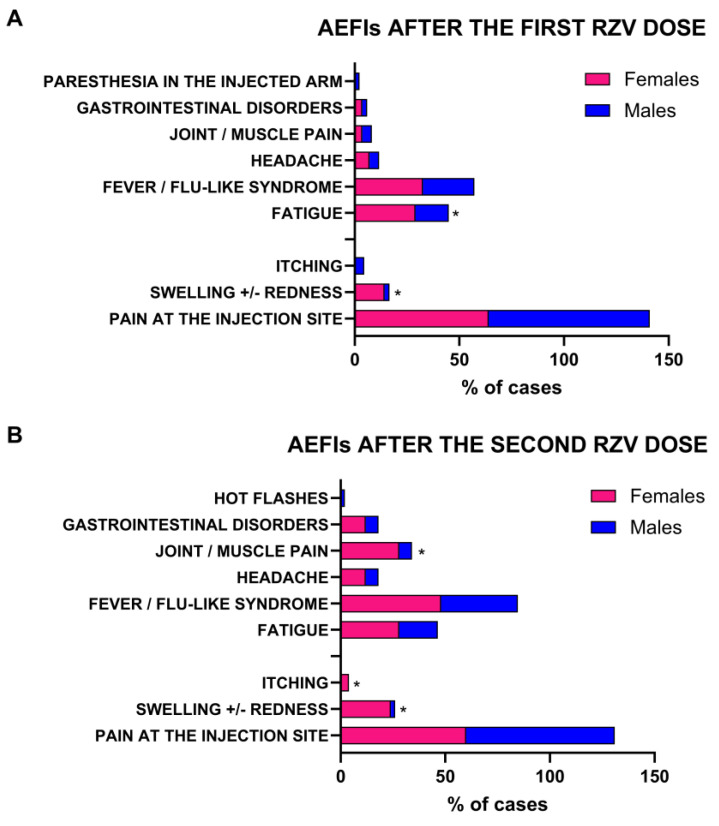
Incidence of adverse events following immunization (AEFI) stratified by sex after first and second recombinant zoster vaccine (RZV). *, *p* value < 0.05.

**Table 1 pathogens-14-00195-t001:** Patients’ characteristics at enrollment.

Characteristics	Total Cohort
	N = 174
Age, years	
Mean ± SD	57 ± 11.8
Median [range]	57 [28–84]
M/F	123 (71%)/51 (29%)
Comorbidities, n (%)	
Kidney transplant	126 (72)
Liver transplant	14 (8.0)
HIV infection	13 (7.5)
Ulcerative colitis	4 (2.3)
Dialysis	4 (2.3)
Obesity	2 (1.1)
Autoimmune rheumatic diseases	3 (1.7)
Type 2 diabetes	1 (1)
Hematological malignancies	2 (1.1)
Geriatric frailty (age ≥ 65 years), n (%)	5 (3.0)

Abbreviations: HIV, human immunodeficiency virus.

**Table 2 pathogens-14-00195-t002:** Incidence and duration of adverse events following immunization (AEFI) stratified by sex.

Characteristics	Females	Males	*p* Value
	N = 51	N = 123	
Age, years			1.0
Mean ± SD	57.0 ± 12.5	56.9 ± 11.5	
Median [range]	58 [28–78]	57 [33–84]	
Comorbidities, n (%)			
Kidney transplant	39 (76)	87 (70.7)	0.4
Liver transplant	3 (6)	11 (9)	0.3
HIV infection	0 (0)	13 (10.6)	0.02
Ulcerative colitis	3 (6)	1 (0.8)	0.04
Dialysis	2 (4)	2 (1.6)	0.4
Obesity	1 (2)	1 (0.8)	0.5
Autoimmune rheumatic diseases	2 (4)	1 (0.8)	0.2
Type 2 diabetes	1 (2)	0 (0)	0.1
Hematological malignancies	0 (0)	2 (1.6)	0.4
Geriatric frailty (Age ≥ 65 years)	0 (0)	5 (4.1)	0.1
AEFIs after the first dose, n (%)	28 (55)	44 (35.8)	0.02
Total duration (h), mean ± SD	75 ± 74.8	66 ± 114.8	0.7
Local symptom duration (h), mean ± SD (n)	93 ± 81.1 (20)	76 ± 126.7 (35)	0.6
Systemic symptom duration (h), mean ± SD (n)	62 ± 91.7 (14)	50 ± 66.5 (19)	0.7
Medications, n (%)	6 (21.4)	4 (9.1)	0.1
AEFIs after the second dose, n (%)	28 (55)	50 (41)	0.09
Total duration (h), mean ± SD	51 ± 40.4	48 ± 42.7	0.8
Local symptom duration (h), mean ± SD (n)	60 ± 43.8 (19)	47 ± 35.7 (37)	0.2
Systemic symptom duration (h), mean ± SD (n)	41 ± 35.6 (19)	45 ± 47.4 (26)	0.6
Medications, n (%)	8 (29)	13 (26)	0.8

Abbreviations: HIV, human immunodeficiency virus.

**Table 3 pathogens-14-00195-t003:** Incidence, duration, and manifestations of AEFIs stratified by age.

Characteristics	Young Adults	Adults	Older Adults	*p* Value
	28–39 yo	40–65 yo	>65 yo	
	N = 13	N = 115	N = 46	
AEFIs after the first dose, n (%)	7 (54)	51 (44)	14 (30)	>0.05
Total duration (h), mean ± SD	84 ± 46.5	67 ± 111	70 ± 84.4	
Local symptom duration (h), mean ± SD (n)	88 ± 49.6 (6)	82 ± 125 (38)	78 ± 92.9 (11)	
Systemic symptom duration (h), mean ± SD (n)	84 ± 31.7 (3)	42 ± 69 (25)	95 ± 106 (6)	
Medications, n (%)	1 (14)	8 (16)	1 (7)	
Local AEFIs, n (%)	7 (54)	40 (35)	12 (26)	0.11
Pain at the injection site	4 (57)	38 (75)	10 (71)	0.35
Swelling +/− redness	3 (23)	1 (1)	1 (2)	0.004
Itching	0 (0)	1 (1)	1 (2)	0.56
Systemic AEFIs, n (%)	5 (38)	31 (27)	8 (17)	0.21
Fever/Flu-like syndrome	2 (29)	15 (30)	2 (14)	0.23
Joint/muscle pain	1 (14)	1 (2)	1 (7)	0.13
Fatigue	1 (14)	11 (22)	3 (21)	0.91
Headache	1 (14)	2 (4)	1 (7)	0.32
Gastrointestinal disorders	0 (0)	2 (4)	0 (0)	1.0
Paresthesia in the injected arm	0 (0)	0 (0)	1 (7)	0.34
AEFIs after the second dose, n (%)	9 (69)	52 (45)	17 (37)	0.04 #
Total duration (h), mean ± SD	61 ± 48.7	43 ± 34.1	60 ± 55.7	all
Local symptom duration (h), mean ± SD (n)	70 ± 33.5 (7)	46 ± 38.3 (37)	48 ± 19.6 (10)	*p* > 0.05
Systemic symptom duration (h), mean ± SD (n)	47 ± 60.7 (6)	39 ± 29.8 (29)	54 ± 61.5 (10)	
Medications, n (%)	3 (33)	17 (33)	2 (12)	
Local AEFIs, n (%)	7 (54)	42 (37)	12 (26)	0.11
Pain at the injection site	6 (67)	36 (69)	11 (65)	0.27
Swelling +/− redness	1 (11)	5 (10)	1 (6)	0.43
Itching	0 (0)	1 (2)	0 (0)	1.0
Edema	1 (11)	0 (0)	0 (0)	0.07
Systemic AEFIs, n (%)	9 (69)	48 (42)	17 (37)	0.04 #
Fever/Flu-like syndrome	5 (55)	22 (42)	5 (30)	0.04
Joint/muscle pain	0 (0)	8 (15)	3 (18)	1.0
Fatigue	1 (11)	9 (17)	7 (41)	0.3
Headache	1 (11)	5 (10)	0 (0)	0.2
Gastrointestinal disorders	1 (11)	4 (8)	1 (6)	0.6
Hot flashes	0 (0)	0 (0)	1 (6)	0.34

#, Chi square test between young adults and elderly. Abbreviations: AEFIs, adverse events following immunizations.

**Table 4 pathogens-14-00195-t004:** Incidence, duration, and manifestations in patients with AEFIs after first and second RZV dose (Group I/II AEFI) or only after second administration (Group II AEFI).

Characteristics	I/II AEFI	II AEFI	*p* Value
	N = 49	N = 31	
Age, years			0.5
Mean ± SD	56 ± 12.1	54 ± 13.3	
Median [range]	56 [28–81]	50 [33–82]	
M/F	18 (37)/31 (63)	11 (35)/20 (65)	0.9
Local AEFIs, n (%)	34 (69)	27 (87)	0.07
Pain at the injection site	30 (61)	24 (77)	0.1
Swelling +/− redness	4 (8)	2 (6)	1.0
Itching	0 (0)	1 (3)	0.2
Systemic AEFIs, n (%)	47 (96)	26 (84)	0.06
Fever/Flu-like syndrome	20 (41)	12 (38)	1.0
Joint/muscle pain	7 (14)	4 (13)	1.0
Fatigue	13 (27)	4 (13)	0.1
Headache	4 (8)	2 (6)	0.8
Gastrointestinal disorders	3 (6)	3 (10)	0.6
Hot flashes	0 (0)	1 (3)	0.2
Total duration (h), mean ± SD	44 ± 35.2	57 ± 48.7	0.2
Local symptom duration (h), mean ± SD (n)	47 ± 35.4 (32)	56 ± 41.5 (26)	0.4
Systemic symptom duration (h), mean ± SD (n)	38 ± 32.7 (30)	53 ± 57.4 (15)	0.3
Medications, n (%)	17 (35)	5 (16)	0.07

Abbreviations: AEFIs, adverse events following immunizations.

**Table 5 pathogens-14-00195-t005:** Incidence, duration, and manifestations stratified by clinical history of Herpes Zoster (HZ) infection.

Characteristics	No Previous HZ	Previous HZ	*p* Value
	N = 136	N = 38	
AEFIs after the first dose, n (%)	55 (40)	17 (45)	0.6
Total duration (h), mean ± SD	78 ± 111.9	42 ± 38.1	0.2
Local symptom duration (h), mean ± SD (n)	90 ± 122.1 (44)	52 ± 43.5 (11)	0.3
Systemic symptom duration (h), mean ± SD (n)	66 ± 85.6 (25)	26 ± 18.6 (9)	0.2
Medications, n (%)	7 (13)	3 (18)	0.6
Local AEFIs, n (%)	45 (82)	13 (76)	0.6
Pain at the injection site	40 (73)	11 (65)	0.5
Swelling +/− redness	4 (8)	1 (6)	1.0
Itching	1 (2)	1 (6)	0.4
Systemic AEFIs, n (%)	31 (56)	14 (82)	0.05
Fever/Flu-like syndrome	14 (26)	6 (25)	0.4
Joint/muscle pain	2 (4)	1 (6)	0.5
Fatigue	11 (20)	4 (24)	0.8
Headache	2 (4)	2 (12)	0.2
Gastrointestinal disorders	1 (2)	1 (6)	0.4
Paresthesia in the injected arm	1 (2)	0 (0)	0.6
AEFIs after the second dose, n (%)	52 (38)	26 (68)	0.001
Total duration (h), mean ± SD	43 ± 34.8	61 ± 51.1	0.07
Local symptom duration (h), mean ± SD (n)	46 ± 34.0 (36)	62 ± 44.9 (20)	0.1
Systemic symptom duration (h), mean ± SD (n)	42 ± 42.1 (28)	45 ± 44.1 (17)	0.8
Medications, n (%)	16 (31)	6 (23)	0.5
Local AEFIs, n (%)	38 (73)	23 (88)	0.1
Pain at the injection site	33 (63)	20 (77)	0.2
Swelling +/− Redness	5 (10)	2 (8)	0.6
Itching	0 (0)	1 (4)	0.2
Systemic AEFIs, n (%)	48 (92)	26 (100)	0.1
Fever/Flu-like syndrome	21 (40)	12 (46)	1.0
Joint/muscle pain	7 (14)	4 (16)	0.3
Fatigue	12 (23)	5 (19)	0.5
Headache	4 (8)	2 (8)	0.7
Gastrointestinal disorders	4 (8)	2 (8)	1.0
Hot flashes	0 (0)	1 (4)	0.2

Abbreviations: AEFIs, adverse events following immunizations.

**Table 6 pathogens-14-00195-t006:** Multivariate logistic regression analysis.

**Dependent Variable = AEFI After First Dose**				
**Variable**	**Odds Ratio**	**SE**	**95%CI**	***p* Value**
Intercept	1.192	0.8156	−1.428 to 1.785	0.8299
Age (years)	0.983	0.01435	−0.0456 to 0.011	0.2381
Sex [F]	2.163	0.3784	0.0334 to 1.524	0.0415
Comorbidities [Others]	4.375	0.6313	0.3014 to 2.835	0.0194
Comorbidities [HIV infection]	1.530	0.6096	−0.8093 to 1.625	0.4851
Comorbidities [Age > 65 years]	0.716	1.170	−3.361 to 1.703	0.7754
History of HZ infection	0.844	0.5211	−1.224 to 0.8369	0.7454
Number of previous HZ infections	1.881	0.5231	−0.3712 to 1.713	0.2273
**Dependent Variable = AEFI After Second Dose**				
**Variable**	**Odds Ratio**	**SE**	**95%CI**	***p* Value**
Intercept	2.022	0.9145	−1.082 to 2.522	0.4412
Age (years)	0.966	0.0161	−0.0673 to −0.004	0.0300
Sex [F]	1.399	0.4239	−0.5017 to 1.169	0.4279
Comorbidities [Others]	0.866	0.6381	−1.419 to 1.108	0.8223
Comorbidities [HIV infection]	1.898	0.7087	−0.7481 to 2.073	0.3660
Comorbidities [Age > 65 years]	0.935	1.247	−3.190 to 2.119	0.9570
History of HZ infection	3.304	0.5755	0.0873 to 2.364	0.0378
Number of previous HZ infections	1.501	0.6167	−0.7537 to 1.705	0.5100
AEFI after first dose	4.302	0.4317	0.6228 to 2.323	0.0007
Systemic AEFI after first dose (h)	1.015	0.0103	0.001 to 0.042	0.1354
Local AEFI after first dose (h)	0.998	0.0033	−0.011 to 0.003	0.4735

Abbreviations: AEFI, adverse events after immunization; SE, standard error; CI, confidential interval; HIV, human immunodeficiency virus; HZ, Herpes Zoster virus.

## Data Availability

Data are contained within this article.

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
