# Peer review of "Sex, Age, and Previous Herpes Zoster Infection Role on Adverse Events Following Immunization with Adjuvanted Recombinant Vaccine"

_pathogens, 2025, doi:10.3390/pathogens14020195_

Round 1
Reviewer 1 Report
Comments and Suggestions for Authors
It is pleasure to review this manuscript about the adverse events following vaccination of the RZV. The author descript the different factor related with adverse events, and comprehensively analysis the impact of sex, age, and previous Herpes Zoster infection on AEFIs in a large subject. This retrospective real-life study presents mangy factors could play an important role on AEFIs in clinical, and imply AEFIs could be avoided by tailor vaccine administration for vulnerable populations.
The manuscript provides some perspectives to the vaccine design for different group. However, There are some problems, which must be solved before it is considered for publication.
1) In the section of result, the serial number of the title is incorrect.
2) The conclusions needs more in it, author could analysis those factors how to influence the AEFIs on immunity.
Author Response
Comments and Suggestions for Authors
It is pleasure to review this manuscript about the adverse events following vaccination of the RZV. The author descript the different factor related with adverse events, and comprehensively analysis the impact of sex, age, and previous Herpes Zoster infection on AEFIs in a large subject. This retrospective real-life study presents mangy factors could play an important role on AEFIs in clinical, and imply AEFIs could be avoided by tailor vaccine administration for vulnerable populations.
The manuscript provides some perspectives to the vaccine design for different group. However, there are some problems, which must be solved before it is considered for publication.
1)In the section of result, the serial number of the title is incorrect.
2)The conclusions needs more in it, author could analysis those factors how to influence the AEFIs on immunity.
Response to Comments. We really thank this Reviewer for the time you have dedicated to our manuscript and for your valuable comments and suggestions.
For serial number of the title in the Results section, we apologize for this inconsistency, and we have revised the subheads accordingly.
Regarding the Conclusion section, we have revised the section as follows.
“In conclusion, incidence and characteristics of AEFIs after RZV can be influenced by several factors, such as sex, age, and previous HZ infection. Indeed, sex-related differences in immune responses are caused by hormone effects on immune cell maturation and distribution, favoring innate responses and inflammation in males whilst adaptive immunity in females. This physiological diversity is associated with a higher risk of severe infectious diseases in males compared to females, while they show a better response to vaccination with lower incidence of AEFIs. Moreover, age also modifies immune system composition and activation, as immunosenescence frequently occurs with ageing, making immune responses less efficient. In our study, we confirmed that sex and age are principal risk factors of AEFIs in females also after RZV, and younger subjects have a less pronounce immune system activation compared to the elderly. In addition, individuals with a history of HZ infection could exhibit altered immune responses due to prior sensitization, which could influence severity and frequency of AEFIs. Our study emphasizes the importance of considering sex, age, and clinical history when planning vaccine schedule, dose, and monitoring, particularly in vulnerable populations. Differences in incidence and types of AEFIs between sexes and age groups highlights the need to tailor vaccine administration, to optimize its efficacy and minimize discomfort and side effects. A targeted focus on frail populations, for whom currently available data are limited, could significantly contribute to improve vaccination planning and AEFI management. Future studies on larger cohorts stratified by clinical and demographic characteristics will be crucial to confirm our observations and to identify potential predictive factors of AEFI occurrence.”
Reviewer 2 Report
Comments and Suggestions for Authors
I was invited to revise the paper entitled "Sex, Age, and previous Herpes Zoster Infection role on Adverse Events Following Immunization with Adjuvanted Recombinant Vaccine".
It was a retrospective cohort study aimed to evaluate incidence and types of AEFIs following RVZ vaccination.
The study was interesing and focused on an important topic for public health.
Obserbvations:
- Introduction section, lacks in some important information. Authors should describe the impact of hz on fragile patients in terms of incidence and hospitalizations. In addition, Authors should add information on vaccination schedule proposed in the Italian context during study period. The vaccine was free of charge?
- Authors did not reported the study period;
- Which is the setting of the vaccine administration? it is unknown;
- Authors should clarify how patients with previous zoster were considered. Which is the distance from the last HZ episode?
- Authors should perform a multiple comparison correction;
- The main limitation of the study was the lack of a control group (patients without comorbidities). It should be added to limitation section;
- It is unknown if enrolled patients received only HZ vaccine or other vaccination was co-administred. The co-administration of vaccination, frequently amonf fragile patients, can bias study results.
- Did patients were admitted to hospital during followup?
- Among discussion, Authors should compare their results with similar studies from other settings;
- Statistical analysis was poor. Authors should perform a regression model aimed to evaluate factors associated to adverse event.
Author Response
Comments and Suggestions for Authors
I was invited to revise the paper entitled "Sex, Age, and previous Herpes Zoster Infection role on Adverse Events Following Immunization with Adjuvanted Recombinant Vaccine". It was a retrospective cohort study aimed to evaluate incidence and types of AEFIs following RVZ vaccination. The study was interesting and focused on an important topic for public health.
Observations:
Comment 1. Introduction section lacks in some important information. Authors should describe the impact of hz on fragile patients in terms of incidence and hospitalizations.
Response to Comment 1. We thank the Reviewer for this important point, and we have added missing information as follows.
On page 2, lines 74-80, the following text was added “In elderly, HZ infection is frequently associated with high rate of hospitalization, up to 6-fold compared to younger subjects (up to 25 hospitalizations / 100,000 patient-years), and an immunocompromised status can increase more this rate, with an in-patient mortality up to two-fold [PMID: 39505070]. For example, diabetic patients have a hospitalization rate of up to 15.9 per 100,000, and those subjects aged 65 or older, with a BMI >30, and poor glycemic control (HbA1c > 8.0 %) are even more at risk also of re-hospitalization (40%) and of post-HZ complications (25% higher risk) [PMID: 38460790].”
Comment 2. In addition, Authors should add information on vaccination schedule proposed in the Italian context during study period. The vaccine was free of charge? Authors did not reported the study period. Which is the setting of the vaccine administration? it is unknown.
Response to Comment 2. We apologize for this missing information, and on page 3, lines 99-101, the following text was added “In this observational real-life study, a total of 174 frail Caucasian adults (mean age, 57±11.8 years; range, 28–84 years) who received RZV at the "Ruggi" hospital site, University Hospital “San Giovanni di Dio e Ruggi d’Aragona”, Salerno, Italy, were included between 2022 and 2024. RZV schedule consisted of two intramuscular doses of 0.5 mL each, with an interval of 2 to 6 months between each administration, as per Italian guidelines and technical datasheet. Vaccination was free of charge as per clinical Italian practice, and was administered at the site hospital Ruggi following the Italian Health Institute’s recommendations.”
Comment 3. Authors should clarify how patients with previous zoster were considered. Which is the distance from the last HZ episode?
Response to Comment 3. We thank the Reviewer for this point, and we have added missing data on page 8, lines 226-227 as follows “However, none of these subjects reported HZ infection in the 12 months before the vaccination.”
Comment 4. Authors should perform a multiple comparison correction.
Response to Comment 4. For Table 3 results, we have performed separate unpaired t-test with each couple of groups.
Comment 5. The main limitation of the study was the lack of a control group (patients without comorbidities). It should be added to limitation section.
Response to Comment 5. We agree with the Reviewer, and we have added the following limitation on page 10, lines 318-319 “and (iii) the lack of a control group without comorbidities and other immunocompromised conditions”.
Comment 6. It is unknown if enrolled patients received only HZ vaccine or other vaccination was co-administred. The co-administration of vaccination, frequently amonf fragile patients, can bias study results.
Response to Comment 6. We have added missing information on page 3, lines 103-104, as follows “No other vaccinations were administered together with RZV, and at least one month was waited between different vaccine doses.”
Comment 7. Did patients were admitted to hospital during followup?
Response to Comment 7. We have added missing information on page 3, lines 104-105, as follows “After vaccination, patients were monitored by periodical out-patients visits.”
Comment 8. Among discussion, Authors should compare their results with similar studies from other settings.
Response to Comment 8. We thank the Reviewer for this valuable feedback, and we have incorporated additional relevant comparisons to provide a broader discussion of our results,
On page 10, lines 296-307, the following text was added “For example, females are at increased risk of injection site and systemic reactions following influenza vaccines compared to males, regardless of age, [37], as well as anaphylaxis, occurring in females in up to 83% of cases after various vaccinations [39], or immediate hypersensitivity after administration of the monovalent 2009 pandemic influenza A (H1N1) vaccine, that is four time common in females, especially in those aged 30-39 years old [40]. Moreover, after at least one dose of AZD1222 vaccine against the Sars-CoV-2, the most common AEFIs are moderate fever (69.4%), muscle aches (68.6%), fatigue/sleepiness (62.5%), body aches (59.4%), headache (58.5%), pain at the injection site (58.3%), and chills (45.7%), and females are more likely to report gastrointestinal symptoms and other AEFIs compared to males [41]. Similarly, in our cohort receiving RZV, the most commonly reported AEFIs were fever, pain at the injection site, and fatigue, especially in females.”
Comment 9. Statistical analysis was poor. Authors should perform a regression model aimed to evaluate factors associated to adverse event.
Response to Comment 9. We thank the Reviewer for this useful comment and we have added a logistic regression analysis on page 9, lines 246-257.
“Finally, we confirmed the effects of these variables on the occurrence of AEFIs after first and second RZV doses by multivariate logistic regression analysis (Table 6). In particular, female sex (p = 0.0415) and the presence of other comorbidities (p = 0.0194), including rheumatological disorders, hematological malignancies, diabetes, or immunological diseases, were significantly associated with the occurrence of AEFIs after the first vaccine dose. Conversely, age (p = 0.0300), clinical history of HZ infection (p = 0.0378) regardless the number of previous reactivations (p = 0.5100), and the occurrence of AEFIs after the first dose (p = 0.0007) were significantly associated with the development of AEFIs after the second RZV administration.
Table 6. Multivariate logistic regression analysis.
Dependent variable = AEFI after first dose |
||||
Variable |
Estimate |
SE |
95%CI |
P value |
Intercept |
0.1753 |
0.8156 |
-1.428 to 1.785 |
0.8299 |
Age (years) |
-0.01693 |
0.01435 |
-0.0456 to 0.011 |
0.2381 |
Sex [F] |
0.7715 |
0.3784 |
0.0334 to 1.524 |
0.0415 |
Comorbidities [Others] |
1.476 |
0.6313 |
0.3014 to 2.835 |
0.0194 |
Comorbidities [HIV infection] |
0.4255 |
0.6096 |
-0.8093 to 1.625 |
0.4851 |
Comorbidities [Age >65 years] |
-0.3338 |
1.170 |
-3.361 to 1.703 |
0.7754 |
History of HZ infection |
-0.1692 |
0.5211 |
-1.224 to 0.8369 |
0.7454 |
Number of previous HZ infections |
0.6316 |
0.5231 |
-0.3712 to 1.713 |
0.2273 |
Dependent variable = AEFI after second dose |
||||
Variable |
Estimate |
SE |
95%CI |
P value |
Intercept |
0.7042 |
0.9145 |
-1.082 to 2.522 |
0.4412 |
Age (years) |
-0.0349 |
0.0161 |
-0.0673 to -0.004 |
0.0300 |
Sex [F] |
0.3361 |
0.4239 |
-0.5017 to 1.169 |
0.4279 |
Comorbidities [Others] |
-0.1433 |
0.6381 |
-1.419 to 1.108 |
0.8223 |
Comorbidities [HIV infection] |
0.6407 |
0.7087 |
-0.7481 to 2.073 |
0.3660 |
Comorbidities [Age >65 years] |
-0.0673 |
1.247 |
-3.190 to 2.119 |
0.9570 |
History of HZ infection |
1.195 |
0.5755 |
0.0873 to 2.364 |
0.0378 |
Number of previous HZ infections |
0.4063 |
0.6167 |
-0.7537 to 1.705 |
0.5100 |
AEFI after first dose |
1.459 |
0.4317 |
0.6228 to 2.323 |
0.0007 |
Systemic AEFI after first dose (h) |
0.0154 |
0.0103 |
0.001 to 0.042 |
0.1354 |
Local AEFI after first dose (h) |
-0.0024 |
0.0033 |
-0.011 to 0.003 |
0.4735 |
Abbreviations. AEFI, adverse events after immunization; SE, standard error; CI, confidential interval; HIV, human immunodeficiency virus; HZ, Herpes Zoster virus.
Round 2
Reviewer 2 Report
Comments and Suggestions for Authors
The paper was improved.
About logistic regression, Authors should present results as OR instead of estimates.
In addition, Authors did not performed the correction for multiple comparisons (bonferroni? false discovery rate?).
Author Response
Comments and Suggestions for Authors
The paper was improved.
Comment 1. About logistic regression, Authors should present results as OR instead of estimates.
Response to Comment 1. We thank the Reviewer for this comment, and we have calculated the OR instead of estimates, as shown below.
Dependent variable = AEFI after first dose |
||||
Variable |
Odds Ratio |
SE |
95%CI |
P value |
Intercept |
1.192 |
0.8156 |
-1.428 to 1.785 |
0.8299 |
Age (years) |
0.983 |
0.01435 |
-0.0456 to 0.011 |
0.2381 |
Sex [F] |
2.163 |
0.3784 |
0.0334 to 1.524 |
0.0415 |
Comorbidities [Others] |
4.375 |
0.6313 |
0.3014 to 2.835 |
0.0194 |
Comorbidities [HIV infection] |
1.530 |
0.6096 |
-0.8093 to 1.625 |
0.4851 |
Comorbidities [Age >65 years] |
0.716 |
1.170 |
-3.361 to 1.703 |
0.7754 |
History of HZ infection |
0.844 |
0.5211 |
-1.224 to 0.8369 |
0.7454 |
Number of previous HZ infections |
1.881 |
0.5231 |
-0.3712 to 1.713 |
0.2273 |
Dependent variable = AEFI after second dose |
||||
Variable |
Odds Ratio |
SE |
95%CI |
P value |
Intercept |
2.022 |
0.9145 |
-1.082 to 2.522 |
0.4412 |
Age (years) |
0.966 |
0.0161 |
-0.0673 to -0.004 |
0.0300 |
Sex [F] |
1.399 |
0.4239 |
-0.5017 to 1.169 |
0.4279 |
Comorbidities [Others] |
0.866 |
0.6381 |
-1.419 to 1.108 |
0.8223 |
Comorbidities [HIV infection] |
1.898 |
0.7087 |
-0.7481 to 2.073 |
0.3660 |
Comorbidities [Age >65 years] |
0.935 |
1.247 |
-3.190 to 2.119 |
0.9570 |
History of HZ infection |
3.304 |
0.5755 |
0.0873 to 2.364 |
0.0378 |
Number of previous HZ infections |
1.501 |
0.6167 |
-0.7537 to 1.705 |
0.5100 |
AEFI after first dose |
4.302 |
0.4317 |
0.6228 to 2.323 |
0.0007 |
Systemic AEFI after first dose (h) |
1.015 |
0.0103 |
0.001 to 0.042 |
0.1354 |
Local AEFI after first dose (h) |
0.998 |
0.0033 |
-0.011 to 0.003 |
0.4735 |
Comment 2. In addition, Authors did not performed the correction for multiple comparisons (bonferroni? false discovery rate?).
Response to Comment 2. Please note that for Table 3, Fisher’s exact test was performed, and only were stated, Chi-square test for specific two group analysis. All other comparisons were single two group comparisons using unpaired t-test, and no multiple comparisons were performed. We have also added the Fisher’s exact test for multiple groups on lines 125-126.